# Predictors of Short Latency Period Exceeding 48 h after Preterm Premature Rupture of Membranes

**DOI:** 10.3390/jcm10010150

**Published:** 2021-01-04

**Authors:** Marion Rouzaire, Marion Corvaisier, Virginie Roumeau, Aurélien Mulliez, Feras Sendy, Amélie Delabaere, Denis Gallot

**Affiliations:** 1Obstetrics and Gynaecology Department, Clermont-Ferrand University Hospital, 63000 Clermont-Ferrand, France; marion.corvaisier@orange.fr (M.C.); feras10d@hotmail.com (F.S.); adelabaere@chu-clermontferrand.fr (A.D.); dgallot@chu-clermontferrand.fr (D.G.); 2Obstetrics and Gynaecology Department, Emile Roux Hospital, 12 boulevard du Dr Chantemesse, 43012 Le Puy-en-Velay, France; virginie.roumeau@ch-lepuy.fr; 3Biostatistics Unit (DRCI) Clermont-Ferrand University Hospital, 63000 Clermont-Ferrand, France; amulliez@chu-clermontferrand.fr; 4“Translational Approach to Epithelial Injury and Repair” Team, Auvergne University, CNRS, Inserm, GReD, 63000 Clermont-Ferrand, France

**Keywords:** preterm premature rupture of membranes, predictive factors, latency, outpatient

## Abstract

Background: Preterm premature rupture of membranes (PPROM) is a complication responsible for a third of preterm births. Clinical management is initially hospital based, but homecare management is possible if patients are clinically stable 48 h after PPROM. This study set out to determine factors that are predictive of short latency (delivery ≤ 7 days) exceeding 48 h after PPROM, enabling estimation of the prevalence of maternal and neonatal complications and comparison of maternal and fetal outcomes between inpatient and outpatient management. Method: This was a monocentric retrospective study conducted between 1 January 2010 and 28 February 2017 on all patients experiencing PPROM at 24 to 34 weeks + 6 days and who gave birth after 48 h. Maternal, obstetric, fetal, and neonatal variables were included in the data collected. The primary endpoint was latency, defined as the number of days between rupture of membranes and delivery. Results: 170 consecutive patients were analyzed. Short latency could be predicted by the need for tocolysis, a cervical length less than 25 mm at admission and the existence of anamnios. Outpatient follow-up was not found to lead to increased maternal morbidity or neonatal mortality. Conclusion: Our study highlights predictive factors of short latency exceeding 48 h after PPROM. Knowledge of these factors may provide justification for outpatient monitoring of patients presenting with a long cervix, absence of need for tocolysis and persistence of amniotic fluid and, thus, no risk factors after 48 h of admission.

## 1. Introduction

Fetal membranes are composed of two histologic layers, the amnion, which is in contact with amniotic fluid, and the chorion, which is in contact with the maternal decidua [1]. They enclose the fetus throughout pregnancy and act as a barrier between fetal and maternal compartments, providing both mechanical and biological protection against external shocks and ascending vaginal flora bacteria, respectively [2,3,4].

Rupture of membranes is a physiological process occurring at the end of pregnancy. A programmed weakening of the paracervical zone, characterized by collagen remodeling and apoptosis combined with uterine contractions that generate stretch and shear forces, leads to the breakage of membranes [5,6]. Rupture that occurs before the beginning of contractions in 10% of pregnancies is referred to as “premature rupture of membranes” (PROM). About 3% of women experience PROM before 37 weeks of gestation, which is called preterm premature rupture of membranes (PPROM) [7,8,9].

This complication is responsible for one-third of preterm births and increases perinatal morbidity and mortality mainly because of the risk of intrauterine infection, which can lead to early neonatal infection, necrotizing enterocolitis and in utero fetal death [10]. Despite advances in prenatal care over the past 30 years, premature rupture of the membrane and preterm birth are still frequent [11,12].

Approximately 50% of women experiencing PPROM (<37 WG) give birth within 24–48 h after the rupture, and 70% to 90% within 7 days [13]. Patients experiencing PPROM require clinical management in a hospital that provides the necessary care for premature newborns [14]. Once the threshold of fetal viability is reached, management of PPROM is initially hospital based and consists of antibiotic prophylaxis and corticosteroids for fetal lung maturation [14]. The main monitoring objectives are the detection and management of maternal and fetal complications, in particular, intrauterine infection.

Homecare management is possible if patients are clinically stable 48 h after PPROM with no clinical or biological signs suggestive of intrauterine infection [14]. The safety of such outpatient management for women with nonthreatening PPROM has been highlighted in several retrospective studies [15,16,17,18]. Homecare inclusion criteria were based on gestational age, absence of chorioamnionitis, clinical stability at least 72 h after PPROM (up to 7 days depending on the studies), cervical dilation and patient home location.

The interval between rupture of the membranes and the onset of labor, referred to as latency, is reported to be associated with neonatal morbidity and mortality [19]. Although some variables, such as gestational age at PPROM, cervical dilatation, parity, twin pregnancy, and chorioamnionitis, have been reported in the literature to affect latency, the time between rupture and delivery remains difficult to predict [20,21,22,23,24]. Knowledge of predictive factors of short latency could optimize the duration of hospital stay and better predict the risk of adverse perinatal outcome. The aim of the present study was to determine predictive factors of short latency exceeding 48 h after PPROM at 24 to 34 weeks + 6 days of gestation.

## 2. Methods

This was a monocentric retrospective study of all pregnant women admitted to the high-risk pregnancy unit of our department between January 2010 and February 2017 with a diagnosis of PPROM (24–34 weeks + 6 days). Obstetric records were identified using O42 codes for premature membrane rupture from the hospital Physician Information System program, including the following:
-0421: Premature rupture of membranes with start of labor after 24 h.-0422: Premature rupture of membranes, labor delayed by treatment.-0429: Premature rupture of membranes, unspecified.

We determined predictive factors of short latency (≤7 days) beyond 48 h after PPROM and estimated both the prevalence of maternal complications (chorioamnionitis based on Newton criteria (clinical and biological variables) and histological chorioamnionitis, postpartum hemorrhage, and postpartum endometritis) and of neonatal complications (low Apgar score, maternal-fetal infection, neonatal intensive care unit admission, and death).

Pediatric records corresponding to the selected mothers were identified via a patient permanent identification number (PIN) and accessed for delivery data using ICOS software (version 1, Auriol, FRANCE), followed by M-EVA for data on hospitalization in a neonatal intensive care unit.

Data included medical history and maternal characteristics, bacteriological tests, ultrasound variables (oligohydramnios was defined as deepest vertical amniotic fluid pocket below 20 mm and anamnios was defined as absence of amniotic fluid), interventions (corticosteroids, antibiotics, and tocolysis), obstetric/neonatal outcomes (delivery, infection, low Apgar score, and NICU admission) and complications that occurred during latency (uterine infection and placental abruption) and delivery. Patients were divided into three groups according to the latency period between PPROM and delivery: less than or equal to 7 days (group 1), between 8 and 14 days (group 2) or more than 14 days (group 3).

All patients received 12 mg betamethasone intramuscularly at admission for PROM and one day later. Antibiotic prophylaxis was administered at admission in monotherapy (third-generation parenteral cephalosporin, 1 g/24 h i.v. for 48 h). Antibiotic therapy was then adapted depending on bacteriological samples collected at admission. Early discontinuation of antibiotic prophylaxis was proposed after 48 h when bacteriological samples were negative [25].

From 7 days of hospitalization, patients we considered stable were offered the possibility of outpatient management if they lived less than 30 min away from the maternity ward. Outpatient management consisted of twice-weekly clinical and fetal monitoring by the midwife (a home visit and a hospital appointment), an ultrasound amniotic fluid assessment and a biological test (complete blood count, C-reactive protein, vaginal and urinary bacteriological samples) once a week.

Statistics were processed using Stata statistical software (version 16, StataCorp, College Station, TX, US). All tests were bilateral, and a *p*-value of <5% was considered statistically significant. The data were described as frequency and associated percentages for categorical data and as mean ± standard deviation for continuous data. Complications and latency analysis were performed using chi-square tests (or Fisher’s exact test when appropriate) for categorical data and using analysis of variance (or Kruskal-Wallis if normality conditions were not met) for continuous data. Normality was assessed graphically and using the Shapiro-Wilk test. A multivariable logistic regression model was used for latency analysis (>7 days vs. <7 days). Covariates were selected according to their clinical relevance and statistical significance (*p* < 0.15) in univariate analysis. Results are shown as odds ratios and 95% confidence intervals. The protocol was approved by our institutional ethical review board (reference: 2020 / CE 96).

## 3. Results

We reviewed and investigated data from 421 patients experiencing PPROM, of whom 187 had a diagnosis of PPROM between 24 weeks and 34 weeks + 6 days, with an expectant management 48 h after rupture. After exclusion of 17 patients for incomplete information, 170 files were analyzed. A patient recruitment flow chart is shown in Figure 1.

Patients were divided into three groups according to latency: less than or equal to 7 days (group 1, *n* = 53), between 8 and 14 days (group 2, *n* = 49) or more than 14 days (group 3, *n* = 68). The groups were similar for maternal characteristics and past obstetrical history (Table 1). History of PPROM for a previous pregnancy did not exceed 10% in each group. The number of twin pregnancies was similar between groups. There was also no difference in intercurrent events occurring during pregnancy including gestational diabetes, pre-eclampsia, bleeding, polyhydramnios and invasive procedures (Table 2).

The mean gestational age at PPROM was 30.54 ± 2.6 weeks. The study demonstrated an association between latency and the proportion of patients with cervical length <25 mm at admission (*p* = 0.02) (Table 3). A patient with a cervical length less than 25 mm was three times more likely to present a latency of ≤7 days (OR = 2.86 (1.39–6.25) 95% CI).

There was no significant association between positive bacteriological samples collected at admission (urinary or vaginal) and latency (*p* = 0.43). In total, 28.2% and 7.6% of patients were found to have a positive vaginal and urinary culture, respectively. At least one bacteriological sample was positive in 31.7% of patients. No difference in white blood cell (WBC) number, CRP and Hemoglobin values at admission was found between the three groups (Table 3).

Oligoamnios was not predictive of short latency, but anamnios rate was significantly associated with a shorter latency period (*p* = 0.001). Anamnios at admission was more frequent in group 1 than in the other groups (*n* = 16 in group 1 vs. *n* = 4 and *n* = 6 patients in groups 2 and 3, respectively) (Table 3). In the same way, the presence of anamnios between day 2 and day 7 of the expectant period was associated with lower latency (*p* = 0.02) (Table 4).

Occurrence of bleeding and the need for antibiotic therapy after positive vaginal swab (VS) or urine culture (UC) were not associated with latency. The need for tocolysis during the first week of the expectant period was associated with short latency (*p* < 0.001) (Table 4). For the latency period of 8 to 14 days, the need for tocolysis was associated with group 2 (*p* = 0.02) (Table 4).

Delivery modalities were the same between the three groups (Table 5). All the patients in groups 1 and 2 delivered before 37 weeks of gestation, while 27.9% of deliveries occurred after 37 weeks in group 3. The mean delivery term was 33.3 ± 3.3 weeks, and the mean latency was 19.4 ± 19 days.

Regarding maternal and neonatal outcomes, we found that Newton-based chorioamnionitis occurred more frequently in group 1 (*p* = 0.02), with nearly one out of two patients affected. In this same group, 54.7% of patients gave birth by cesarean section, all performed as emergencies. Other maternal complications were equal in the three groups. Birth weight was significantly affected by latency (Table 5). No difference was found in the frequency of an Apgar score < 7 at 5 min between groups, but four neonatal deaths occurred in the first group (latency equal to or less than 7 days):
-During delivery due to head entrapment in a breech presentation at 28.3 weeks;-at day 1 due to respiratory failure after cesarean section (C/S) birth in the context of chorioamnionitis at 29 weeks;-at day 1 due to respiratory failure after C/S birth for breech presentation at 25.4 weeks;at day 9 due to sepsis after C/S birth for cord prolapse at 25.1 weeks.

Comparison of inpatient versus outpatient management showed significant differences in variables measured at admission: 50% of hospitalized patients had a cervical length of less than 25 mm versus 17.6% in the outpatient group (*p* < 0.01), and 60.5% of inpatients had uterine contractions requiring tocolysis compared to 31.4% in the outpatient group (*p* = 0.01) (Table 6). Latency duration was significantly lower in hospitalized patients (11.2 ± 10.2 days vs. 38.6 ± 23.1 days, *p* = 0.01). Chorioamnionitis occurred more frequently in the hospitalized group (42%) than in the outpatient group (23.5%) (*p* = 0.02). The frequency of preterm spontaneous labor was significantly lower among outpatients (*p* = 0.001), while induction of labor was more frequent (*p* = 0.01). No home delivery or significant difference in the mean term of delivery were noted. Newborn birth weight was found to be significantly higher in the outpatient group (*p* = 0.01) (Table 6), but no differences were found in maternal-fetal infection rate between the two groups.

## 4. Discussion

In the literature, spontaneous labor occurs within 24 to 48 h after PPROM in approximately 50% of women and 70–90% give birth within 7 days [13,26,27,28]. In our study, 46.4% of women began labor within 48 h but only 62.2% gave birth within 7 days. We excluded patients beyond 34 weeks + 6 days, which may offer an explanation for this difference.

The factors predicting short latency highlighted in our series were initial and persistent anamnios, short cervical length (< 25 mm) at admission, and the need for tocolysis between day 2 and day 7 of the expectant period.

On the other hand, a positive bacteriological sample (VS or UC) did not influence latency in our series, contrary to the results of Dagklis et al. in 2013 [20]. A positive sample was not associated with the occurrence of chorioamnionitis, thus, supporting data reported in the literature [29]. The microbiological profile of chorioamnionitis is generally defined by the positivity of bacterial cultures or PCR performed on amniotic fluid, though the relevance of this approach remains debatable in view of the numerous false negatives (depending on the germs tested and techniques used) and false positives, including in asymptomatic patients [29]. Antibiotic prophylaxis was administered at admission for PPROM and adapted depending on bacteriological samples collected at admission. The benefits of antibiotic administration in prolonging the latency period have already been well documented in the literature [30,31].

In our series, the presence of oligohydramnios was not predictive of latency, but the existence of anamnios was associated with short latency. In the literature, the link between degree of oligohydramnios and latency has previously been highlighted. The more severe the degree of oligohydramnios, the shorter the latency period [13,32,33]. While several studies have demonstrated that anamnios did not affect neonatal prognosis [21,34,35], Hadi et al. [33] found that low amounts of amniotic fluid constituted a prenatal prognosis factor. Melamed et al. [24] also observe that neonatal adverse outcome was more likely in cases of oligohydramnios, In this context, the amnioinfusion technique was used by some teams to extend the latency period, with disappointing results [36].

In our study, most anamnios diagnosed at admission were persistent until the end of pregnancy. Only two cases resulted in oligoamnios during the expectant period. Seven patients had normal amniotic fluid volume at admission becoming anamnios during the first 7 days. The association between completion of anamnios during the first week and latency could not be demonstrated in our series (*p* = 0.052), possibly reflecting the lack of power of the study.

Most variables selected for identifying patients at risk of premature delivery were known within the first 48 h after preterm PROM. At admission, amniotic fluid quantification and ultrasound measurement of cervical length enabled identification of patients at risk of premature delivery. This may allow for outpatient follow-up to be considered for patients with normal amniotic fluid volume and a long cervical length (≥25 mm) at admission. Cervical assessments should be limited regardless of the method used [14]. The emergence of painful uterine contractions that require tocolysis between day 2 and day 7 appears to be an additional risk factor. For the latency period of 8–14 days, an association was found between need for tocolysis and risk of imminent delivery. The need for tocolysis, which is a subjective variable, is in our opinion more relevant than frequency or intensity of uterine contractions, which do not correlate with patient pain.

No placental abruptions were found in our series, but the other identified complication rates were the same as those described in the literature. Our cesarean section rate was, however, higher than rates reported in the literature [13,20,21]. This impaired vaginal/ cesarean balance is likely due to our study design, which excluded patients with a latency of less than 48 h and PPROM beyond 34SA + 6 days.

Although the primary purpose of this study was not to evaluate the feasibility of outpatient follow-up, our results showed this care to be safe and did not seem to lead to maternal-fetal complications, in support of other retrospective studies comparing inpatient and outpatient management of PPROM. In Beckman et al.’s 2013 study [17], the evaluation period was 72 h, and in Garabedian et al. [15] (2015), the average assessment period was 7 days with no significant maternal-fetal complications reported between the two groups. Both studies emphasized the need for well-defined criteria to select patients eligible for outpatient monitoring.

In our series, the rate of patients with cervical length <25 mm, uterine contractions requiring tocolysis, and chorioamnionitis rate were higher in the hospitalized group compared to the outpatient group with an extended latency period in the outpatient group. As outpatient management was only proposed after day 7 for stable patients, the extended latency period in this group reflected a selection bias and was also related to a lower obstetrical risk [15,16].

We found outpatient monitoring to be a safe option for the management of PPROM after 48 h of admission for patients with none of these risk factors, i.e., a long cervix and absence of anamnios. A prospective randomized trial with a larger number of subjects would now be useful to compare inpatient and outpatient care in terms of safety, maternal satisfaction and economic cost [16].

## 5. Conclusions

Our study highlights predictive factors of short latency exceeding 48 h after PPROM. Knowledge of these factors may provide justification for outpatient monitoring of patients presenting with a long cervix, absence of need for tocolysis and persistence of amniotic fluid and, thus, no risk factors after 48 h of admission.

## Figures and Tables

**Figure 1 jcm-10-00150-f001:**
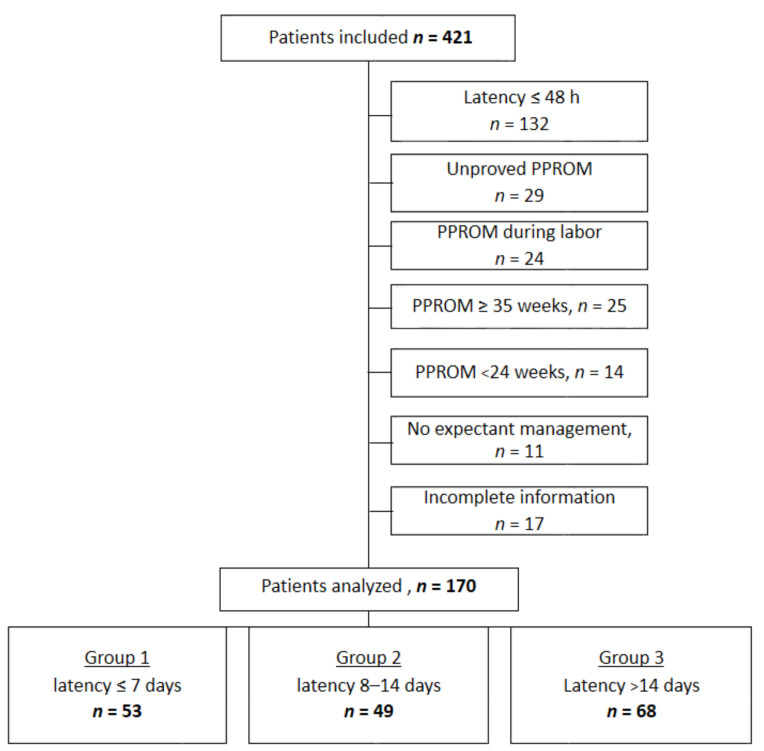
Flowchart.(n: number of subject)

**Table 1 jcm-10-00150-t001:** Maternal characteristics and past obstetrical history.

Variable	Group 1 (*n* = 53)	Group 2 (*n* = 49)	Group 3 (*n* = 68)	Total (*n* = 170)	*p*
Mean maternal age, y, mean ± SD Gravida Parity	29.8 ± 6.6 2.5 ± 1.5 0.8 ± 1.0	30.4 ± 6.1 2.4 ± 1.4 0.7 ± 0.8	29.3 ± 5.6 2.4 ± 1.4 0.9 ± 1.2	29.8 ± 6.0 2.4 ± 1.5 0.8 ± 1.1	0.52 0.77 0.82
Uterine malformation, *n* (%)	0	1 (2.0)	1 (1.5)	2 (1.0)	0.75
Type 1 or 2 diabetes, *n* (%)	1	0	1	2 (1.0)	0.65
BMI, kg/m², mean ± SD	23.4 ± 5.0	23.1 ± 4.0	24.0 ± 5.2	23.5 ± 4.8	0.87
Cervical conization, *n* (%)	3 (5.7)	2 (4.1)	0	5 (2.9)	0.11
Active smoker, *n* (%)	18 (33.9)	21 (42.8)	29 (42.6)	68 (40.0)	0.56
Passive smoker, *n* (%)	9 (17.0)	3 (6.1)	12 (17.6)	24 (14.1)	0.17
Previous termination of pregnancy, *n* (%)	12 (22.6)	9 (18.3)	10 (14.7)	31 (18.2)	0.52
Previous spontaneous abortion, *n* (%)	14 (26.4)	11 (22.4)	12 (17.6)	37 (21.7)	0.48
Previous ectopic pregnancy, *n* (%)	4 (7.5)	3 (6.1)	1 (2.0)	8 (4.7)	0.25
Previous preterm labor, *n* (%)	7 (13.2)	8 (16.3)	12 (17.6)	27 (15.8)	0.72
Previous PPROM, *n* (%)	2 (3.7)	2 (4.8)	6 (8.8)	10 (5.9)	0.41
Previous C/S, *n* (%)	6 (11.3)	7 (14.3)	4 (5.9)	17 (10.0)	0.30

PPROM: preterm premature rupture of membranes; C/S: cesarean section; group 1: latency ≤ 7 days, group 2: latency 8–14 days; group 3: latency > 14 days.

**Table 2 jcm-10-00150-t002:** Characteristics of pregnancy and intercurrent events.

Variable, *n* (%)		Group 1 (*n* = 53)	Group 2 (*n* = 49)	Group 3 (*n* = 68)	Total (*n* = 170)	*p*
Spontaneous pregnancy		50 (94.3)	45 (91.8)	61(89.7)	156	0.65
Pregnancy via ARTs		3 (5.6)	4 (8.1)	7 (10.3)	14	0.64
Single pregnancy		48 (90.6)	44 (89.8)	63(92.6)	155	0.42
Twin pregnancy		5 (9.43)	5 (10.2)	5 (7.3)	15	0.85
Gestational diabetes		8 (15.1)	5 (10.2)	7 (10.3)	20	0.66
Pre-eclampsia		2 (3.8)	1 (2.0)	1 (1.5)	4	0.67
Bleeding *n* (%)		9 (16.9)	17(34.7)	10 (14.7)	36	0.50
	T1	6 (11.3)	5 (10.2)	7 (10.3)	18	0.98
	T2	3 (5.7)	7 (14.3)	2 (3.0)	12	0.06
	T3	0	5 (10.2)	1 (1.5)	6	0.10
Polyhydramnios		4 (7.5)	1 (2.0)	0	5	0.06
Invasive procedure during pregnancy		8 (15.1)	8 (16,3)	4 (5,9)	20	0.19

ART: assisted reproductive technology, group 1: latency ≤ 7 days, group 2: latency 8–14 days; group 3: latency > 14 days.

**Table 3 jcm-10-00150-t003:** Admission data.

Variable		Group 1 (*n* = 53)	Group 2 (*n* = 49)	Group 3 (*n* = 68)	Total (*n* = 170)	*p*
Gestational age at PPROM, w Mean ± SD		30.97 ± 2.6	30.72 ± 2.5	30.08 ± 2.6	30.54 ± 2.6	0.14
Col < 25 mm, *n* (%)		27 (50.9)	21(42.8)	18 (26.5)	66 (38.8)	0.02
Need for tocolysis, *n* (%)		30 (56.6)	24 (49.0)	28 (41.1)	82 (48.2)	0.24
Positive VS, *n* (%)		17 (32.1)	12 (24.5)	18 (26.5)	48 (28.2)	0.70
Positive UC, *n* (%)		4 (7.5)	0	9 (13.2)	13 (7.6)	0.29
Positive vaginal or urinary test *n* (%)		18 (33.9)	12 (24.5)	24 (35.3)	54 (31.7)	0.43
WBC count, G/L Mean ± SD		12.9 ± 4.2	12.7 ± 3.7	12.2 ± 3.8	12.5 ± 3.9	0.47
Hemoglobin, g/dL Mean ± SD		11.3 ± 1.2	11.6 ± 1.3	11.6 ± 1.1	11.5 ± 1.2	0.24
CRP, mg/L Mean ± SD		6.24 ± 4.6	11.64 ± 11.5	5.80 ± 5.8	7.70 ± 8.1	0.35
Amniotic fluid at admission, *n* (%)						
	Normal	26 (49.1)	30 (61.2)	43 (63.2)	99 (58.2)	0.26
	Oligoamnios	11 (20.7)	15 (30.6)	19 (27.9)	45 (26.4)	0.45
	Anamnios	16 (30.2)	4 (8.2)	6 (8.8)	26 (15.3)	0.001

VS: vaginal swab; UC: urine culture; WBC: white blood cells; CRP: C-reactive protein; group 1: latency ≤ 7 days, group 2: latency 8–14 days; group 3: latency > 14 days.

**Table 4 jcm-10-00150-t004:** Expectant period.

Variable			Group 1 (*n* = 53)	Group 2 (*n* = 49)	Group 3 (*n* = 68)	Total (*n* = 170)	*p*
Permanent Hospital care, *n* (%)			52 (98.1)	43 (87.8)	24 (35.3)	119 (70)	< 0.001
Ambulatory care, *n* (%)			1 (1.9)	6 (12.2)	44(64.7)	51 (30)	
Day 2 to day 7							
	Bleeding *n*(%)		7 (13.20)	4 (8.2)	6 (8.8)	17 (10)	0.69
	Need for tocolysis *n* (%)		20 (37.7)	6 (12.2)	6 (8.8)	32 (18.8)	< 0.001
	Amniotic fluid, *n*(%)	Normal	23 (43.4)	23 (46.9)	43 (63.2)	89 (52.3)	
		Oligohydramnios	12 (22.6)	16 (32.6)	16 (23.5)	44 (25.9)	0.47
		Anamnios	18 (33.9)	10 (20.4)	9(13.2)	37 (21.8)	0.02
	Antibiotic therapy after VS+ or UC+, *n* (%)		14 (26.4)	14 (28.6)	20 (29.4)	48 (28.2)	0.93
Day 8 to day 14							
	Bleeding *n* (%)		NA	9 missing items of data 2 (5)	35 missing items of data 3 (9.1)	5 (6.6)	0.72
	Need for tocolysis *n* (%)		NA	9 missing items of data 10 (25.0)	35 missing items of data 1 (3.0)	11 (14.5)	0.02
	Amniotic fluid, *n*(%)			10 missing items of data	36 missing items of data		0.42
		Normal	NA	10 (25.6)	12 (37.5)	22 (29.7)	
		Oligoamnios	NA	18 (46.1)	10 (31.2)	29 (39.2)	
		Anamnios	NA	11 (28.2)	10 (31.2)	23 (31.1)	
	Antibiotic therapy after VS+ or UC+, *n* (%)		NA	9 missing items of data 6 (15)	36 missing items of data 14 (43.7)	21 (28)	0.02

VS: vaginal swab; UC: urine culture; NA: not applicable; group 1: latency ≤ 7 days, group 2: latency 8–14 days; group 3: latency > 14 days.

**Table 5 jcm-10-00150-t005:** Delivery modalities and neonatal outcomes.

Variable		Group 1 (*n* = 53)	Group 2 (*n* = 49)	Group 3 (*n* = 68)	Total (*n* = 170)	*p*
Term, w Mean (± SD)		31.6 ± 2.6	32.3 ± 2.6	35.3 ± 3.1	33.3 ± 3.3	0.38
< 37 weeks, *n* (%)		53 (100)	49 (100)	49 (72.0)	151 (88.9)	
> 37 weeks, *n* (%)		0	0	19 (27.9)	19 (11.1)	
Latency, d Median (IQR)		5 (4–6)	10 (9–12)	29 (21–52)		
Spontaneous labor, *n* (%)		33 (62.2)	30 (61.2)	31 (45.6)	94 (55.3)	0.12
Induction of labor *n* (%)		1 (1.9)	3 (6.1)	20 (29.4)	24 (14.1)	0.50
Presentation, *n* (%)						
	Cephalic	39 (73.6)	44 (89.8)	53 (61.8)	136 (80)	0.09
	Others	15 (28.3)	5 (10.2)	14 (4,1)	34 (20)	0.06
Deliveries, *n* (%)						
	Spontaneous	22 (41.5)	22(44.9)	42 (61.8)	86	0.08
	Instrumental	2 (3.8)	4 (8.2)	3 (4.4)	9	0.62
	Planned C/S	0	1 (2.0)	5 (7.3)	6	0.09
	Emergency C/S	29 (54.7)	22 (44.9)	18 (26.5)	69	0.06
Maternal complications, *n* (%)						
	Clinical-biological CA	26(49.0)	17(34.7)	17(23.5)	60	0.02
	Histological CA	21(39.6)	18(36.7)	16(23.5)	55	0.13
	Endometritis	6(11.3)	3(6.1)	5(7.3)	14	0.60
	Placental retention	2(3.8)	2 (4.1)	2 (2.9)	6	0.95
	PPH	7 (13.2)	6 (12.2)	9 (13.2)	22	0.98
	Placental abruption	0	0	0	0	-
	Cord prolapse	1 (1.9)	1 (2,9)	2 (2,9)	4	0.92
Neonatal outcomes						
	Birth weight, g Mean ±SD	1652 ±516	1793 ±518	2416 ±641	1993 ±663	0.001
	Apgar < 7 at 5 min *n* (%)	9 (17)	11 (22.4)	14 (69.3)	34	0.43
	Maternal-fetal infection	16 (30.2)	16 (32.6)	12 (62.3)	44	0.16
	Neonatal death	4 (7.5)	0	0	4	0.77

C/S: cesarean section; CA: chorioamnionitis; PPH: postpartum hemorrhage; IQR: interquartile range; group 1: latency ≤ 7 days, group 2: latency 8–14 days; group 3: latency > 14 days.

**Table 6 jcm-10-00150-t006:** Inpatient and outpatient management.

Variable	Inpatients *n* = 119	Outpatients *n* = 51	Total = 170	*p*
Admission information				
Cervix < 25 mm, *n* (%)	57(47.9)	9 (17.7)	66 (38.8)	< 0.001
Need for tocolysis, *n* (%)	66 (55.5)	16 (31.4)	82 (48.2)	0.004
Amniotic fluid, *n* (%)				0.001
Normal	59 (49.6)	40 (78.4)	99 (58.2)	
Oligoamnios	35 (29.4)	10 (19.6)	45 (26.5)	
Anamnios	25 (21.0)	1 (2.0)	26 (15.3)	
Delivery modalities and complications, *n* (%)				
Latency, d, mean ±SD	11.2 ± 10.2	38.6 ± 23.1	19.4 ± 23.1	0.01
Spontaneous labor	72 (63.1)	22 (43.4)	94	0.02
Induction of labor	5 (4.4)	19 (37.2)	24	0.01
Spontaneous vaginal delivery	52 (45.6)	33 (64.7)	85 (50)	0.01
Instrumental delivery	7 (5.9)	2 (3.9)	9 (5.3)	0.72
Planned C/S	2 (1.7)	4 (7.8)	6 (3.5)	0.07
Emergency C/S	58 (48.7)	11 (21.6)	69 (40.6)	0.001
Chorioamnionitis				
Clinical and biological	48(40.3)	12 (23.5)	60 (35.3)	0.04
Histological	46 (37.0)	9 (17.6)	55 (33.1)	0.01
Gestational age at delivery, weeks, Mean ± SD	31.8 ± 3.9	36.2 ± 2.5	33.1 ± 4.1	< 0.001
Neonatal information				
Birth weight, g Mean ± SD	1743 ± 535	2583 ±558	1993 ± 663	< 0.001

Neonatal infection	35 (29.4)	9 (17.6)	44 (25.9)	0.126

Multivariate analysis demonstrated higher risk for latency ≤ 7 days with initial and persistent anamnios (OR = 6.73 (2.49–18.17) 95% CI), need for tocolysis between day 2 and day 7 (OR = 4.57 (1.97–10.62) 95% CI), cervix ≤ 25 mm (OR = 2.80 (1.49–5.28) 95% CI), and advanced gestational age (OR = 1.23 (1.08–1.40) 95% CI).

## Data Availability

The data presented in this study are available on request from the corresponding author. The data are not publicly available due to privacy.

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
