# Peer review of "Predictors of Short Latency Period Exceeding 48 h after Preterm Premature Rupture of Membranes"

_jcm, 2021, doi:10.3390/jcm10010150_

Round 1
Reviewer 1 Report
Data reported by the authors are very interesting and not easily reproducible because the practice of home management of pPROM and of conservative management after 34 and even after 37 weeks is uncommon. Ethical approval of this protocol is available?
Methods and results
The main outcome is the latency or the definition of the factors affecting the latency? The abstract should be more precise.
The use of tocolysis in the context of pPROM is controversial. Why no mention of corticosteroids administration?
Which were the criteria for home management? The authors report (in the conclusion) that the same factors observed as related to longer latencies were those used to decide home management. This information, even if clinically sound, could constitute a bias because were managed at home cases more favorable. Then we cannot derive from this paper the effect of home management and the relative table 6 could suggest an erroneous message.
The definition of ologohydramnios should be reported.
Table 4 is not clear. A graphic representation of some events during latency could be more readable
Author Response
Data reported by the authors are very interesting and not easily reproducible because the practice of home management of pPROM and of conservative management after 34 and even after 37 weeks is uncommon. Ethical approval of this protocol is available?
- We added the reference of the ethical approval (2020 / CE 95) in the method part (line 117). : « The protocol was approved by our institutional ethical review board (reference: 2020 / CE 95).”
Methods and results
The main outcome is the latency or the definition of the factors affecting the latency?
- Our main outcome was the latency ie the number of days between rupture of the membranes and onset of labor
The abstract should be more precise.
- We clearly mentionned that “…This study set out to determine factors that are predictive of short latency (delivery ≤ 7 days) exceeding 48 h after PPROM… »
The use of tocolysis in the context of pPROM is controversial. Why no mention of corticosteroids administration?
- We added: “All patients received betamethasone 12mg intra-muscularly at admission for PROM and one day later. » (line98-99) Sorry for this omission.
- Concerning tocolysis, our national guidelines consider that « In the absence of any demonstrated neonatal benefits, there is not sufficient evidence to recommend (or to recommend against) initial tocolysis for preterm PROM (Grade C). If tocolysis is prescribed, it should not continue longer than 48 h (Grade C).”(Schmitz et al., 2019 -Eur J Obstet Gynecol Reprod Biol)
Which were the criteria for home management?
- From 7 days of hospitalization, patients considered stable were offered the possibility of outpatient management.The elements that allowed us to authorize the patient's discharge were :
- Adherence to extra-hospital supervision.
- The possibility of rest at home (good socio-economic conditions).
- A place of residence less than 30 minutes away from the maternity ward.
- The absence of clinical or paraclinical signs of intrauterine infection.
- The absence of uterine contractions.
- The absence of abundant amniotic fluid loss.
- Good fetal vitality.
- Absence of anamnios on ultrasound.
- Normal biological parameters.
- Negative bacteriological samples.
- We added: “line(103-104) “From 7 days of hospitalization, patients we considered stable were offered the possibility of outpatient management if they lived less than 30 minutes away from the maternity ward. »
The authors report (in the conclusion) that the same factors observed as related to longer latencies were those used to decide home management. This information, even if clinically sound, could constitute a bias because were managed at home cases more favorable. Then we cannot derive from this paper the effect of home management and the relative table 6 could suggest an erroneous message.
- We clearly discussed the selection bias in the outpatient group (Line 251-254)
. « As outpatient management was only proposed after day 7 for stable patients, the extended latency period in this group reflected a selection bias and was also related to a lower obstetrical risk »
The definition of oligohydramnios should be reported.
- We added the definition of oligohydramnios and anamnios in the methods part (line 91-92)
- « Data included medical history and maternal characteristics, bacteriological tests, ultrasound variables (oligohydramnios was defined as deepest vertical amniotic fluid pocket below 20mm and anamnios was defined as absence of amniotic fluid),… »
Table 4 is not clear. A graphic representation of some events during latency could be more readable
- We finally considered that Table 4 could not be easily replaced by a single graphic representation due to the amount of data and the difficulty to declare missing data appropriately.
Submission Date
19 November 2020
Date of this review
03 Dec 2020 15:07:48

Reviewer 2 Report
It is a retrospective study related to the predictors of short latency beyond the 48 hours after PPROM.
The present study is of adequate scientific quality and trial’s design.
- Introduction is clear and with right length. However, at the scopes of the trial, could be enclosed the inpatient and the outpatients outcomes of the management provided in the study, since that the former is the cornerstone of the study.
- In Methodology must be clarified why the author selected the three different group of the study, instead of two groups (>7days<). Institutional review BOARD, which approved the present study, was not mentioned.
- Results are presented good enough.
- Discussion is clear and informative. More comments are needed concerning the table 6, to improve the clinical significance of the manuscript as well as to highlight the conclusion of the study.
Author Response
Comments and Suggestions for Authors
It is a retrospective study related to the predictors of short latency beyond the 48 hours after PPROM.
The present study is of adequate scientific quality and trial’s design.
- Introduction is clear and with right length. However, at the scopes of the trial, could be enclosed the inpatient and the outpatients outcomes of the management provided in the study, since that the former is the cornerstone of the study.
The elements that allowed us to authorize outpatient management were :
- Adherence to extra-hospital supervision.
- The possibility of rest at home (good socio-economic conditions).
- A place of residence less than 30 minutes away from the maternity ward.
- The absence of clinical or paraclinical signs of intrauterine infection.
- The absence of uterine contractions.
- The absence of abundant amniotic fluid loss.
- Good fetal vitality.
- Absence of anamnios on ultrasound.
- Normal biological parameters.
- Negative bacteriological samples.
- We did not alter the introduction part but we mentionned more clearly how was selected the outpatient population in the methods part (“103-104”)
- We added: “From 7 days of hospitalization, patients we considered stable were offered the possibility of outpatient management if they lived less than 30 minutes away from the maternity ward. »
- In Methodology must be clarified why the author selected the three different group of the study, instead of two groups (>7days<). Institutional review BOARD, which approved the present study, was not mentioned.
- In a preliminary study we observed that duration of latency was approximatively 1/3 during the first week, 1/3 during the second week and 1/3 later. This encouraged to opt for this description rather than to combine all patients delivering later than 7 days in one single group. This option enabled to describe more accurately the prolonged latency after 7 days by describing group 2 (7-14 days) and group 3 (> 14 days). For example, concerning outpatients who were all (except one) selected after 7 days, only 12% delivered during the subsequent week and the vast majority delivered later than 14 days.
- We added the reference of the ethical approval (2020 / CE 95) in the method part (line 117). : « The protocol was approved by our institutional ethical review board (reference: 2020 / CE 95).”
- Results are presented good enough.
- Discussion is clear and informative. More comments are needed concerning the table 6, to improve the clinical significance of the manuscript as well as to highlight the conclusion of the study.
- We clearly discussed the selection bias of the inpatient and outpatient groups (Line 251-254) to improve the clinical significance of these results.
. « As outpatient management was only proposed after day 7 for stable patients, the extended latency period in this group reflected a selection bias and was also related to a lower obstetrical risk »
Submission Date
19 November 2020
Date of this review
06 Dec 2020 22:21:40

Reviewer 3 Report
I read with interest the manuscript entitled: “Predictors of short latency period exceeding 48 hours after preterm premature rupture of membranes” submitted for publications by
Rouzaire et al.
This was a retrospective analysis of 170 women with PPROM, i.e., prior to 34+6 gestational weeks, exploring risk factors for short latency among women with undelivered PPROM at 48 hours.
Overall, the study is well designed, nicely presented and well written. I have some concerns and comments for the authors to address
- Please add in the abstract (and methods) the definition of your primary outcome. i.e., short latency
- Please address and compare your data and results to other published risk factors for short latency – cervical dilatation (clinical not just sonographic), gestational age, parity, amniotic fluid volume and fetal growth - some of which you have mentioned and other not.
The references for these risk factors can be found at: Kibel et al. 2016 (Obstetrics and gynecology), Melamed et al. 2011 (AJOG), Melamed et al. 2009 (Maternal Fetal and Neonatal Medicine)
- Please refrain from using “cases” as they are women, patients or participants.
- Please explain the upper limit of 34+6, as most contemporary guideless (i.e. ACOG guidelines) suggest either 34+0W or lately 37+0W, as the upper limit for conservative management.
- Please consider to include placental abruption as one of the outcomes
- Please explain the use of antibiotics which is not in line with common ACOG practice
- Early discontinuation - define “early”
- Also, why not administer antibiotics for a 7-day period as suggested by ACOGs practice bulletin and extensively studied as beneficial
- Why were cephalosporins chosen and not erythromycin (or other macrolides) plus GBS prophylaxis, if indicated, with penicillin?
- Please explain your routine clinical practice on the endpoint of conservative management - how, when and who decides on induction
- Bleeding and preeclampsia, which occur after PPROM, are not baseline characteristics, rather pregnancy outcomes.
- The length and type of antibiotic treatment, which a proven factor for prolonged latency and may not be similar between tour groups should be presented
- Please define “need for tocolysis” as this in not routinely indicated, and may be a relative contraindication” for PPROM. Who gets tocolysis and for how long, and how does it affect home care
Author Response
Comments and Suggestions for Authors
I read with interest the manuscript entitled: “Predictors of short latency period exceeding 48 hours after preterm premature rupture of membranes” submitted for publications by
Rouzaire et al.
This was a retrospective analysis of 170 women with PPROM, i.e., prior to 34+6 gestational weeks, exploring risk factors for short latency among women with undelivered PPROM at 48 hours.
Overall, the study is well designed, nicely presented and well written. I have some concerns and comments for the authors to address
- Please add in the abstract (and methods) the definition of your primary outcome. i.e., short latency
- We added the definition of short latency in the abstract and methods (line82) “…This study set out to determine factors that are predictive of short latency (delivery ≤ 7 days) exceeding 48 h after PPROM… »
- Please address and compare your data and results to other published risk factors for short latency – cervical dilatation (clinical not just sonographic), gestational age, parity, amniotic fluid volume and fetal growth - some of which you have mentioned and other not.
The references for these risk factors can be found at: Kibel et al. 2016 (Obstetrics and gynecology), Melamed et al. 2011 (AJOG), Melamed et al. 2009 (Maternal Fetal and Neonatal Medicine)
- Cervical dilatation and parity were added as risk factors in the introduction part (line67-70) with references : « Although some variables, such as gestational age at PPROM, cervical dilatation, parity, twin pregnancy, and chorioamnionitis have been reported in the literature to affect latency, the time between rupture and delivery remains difficult to predict.20–24 »
- We found no difference concerning gravida and parity between our 3 groups (p=0,82). We added these results in the Table 1 of the revised manuscript.
- Cervical dilatation was not systematically available as our usual management proposed ultrasonographic assessment of the cervix. In accordance with our national guidelines, the combination of ultrasonographic and digital examination of the cervix was avoided. “If a cervical evaluation seems necessary, an examination by speculum or a digital or ultrasound cervical examination can be performed (professional consensus). Cervical assessments should be limited, regardless of the method used (professional consensus).” (Schmitz et al., 2019 -Eur J Obstet Gynecol Reprod Biol)
- We adressed our results to other published data on amniotic fluid volume (Melamed et al, 2011 - Line 219-220)
- Please refrain from using “cases” as they are women, patients or participants.
- We suppressed the term “cases” line 75, 172, 221 and 236 in the revised manuscript.
- Please explain the upper limit of 34+6, as most contemporary guideless (i.e. ACOG guidelines) suggest either 34+0W or lately 37+0W, as the upper limit for conservative management.
- Since 2005, we adopted an expectant management until 34+6W and not 34+0 W to limit neonatal morbidity because we demonstrated that neonatal morbidity was high in case of systematic delivery at 34 W (Accoceberry et al, 2005 – Gynecol Obstet Fertil).
- Please consider to include placental abruption as one of the outcomes
- No case of abruption was observed in this series
- Please explain the use of antibiotics which is not in line with common ACOG practice
- The use of antibiotics reported in this series was in accordance with our national recommendations (Schmitz et al., 2019 -Eur J Obstet Gynecol Reprod Biol)
- Early discontinuation - define “early”
- Early discontinuation refered to discontinuation 48h after admission in case of negative bacteriological samples. We precised this point in the revised manuscript (line 102):
- « Early discontinuation of antibiotic prophylaxis was proposed after 48 h when bacteriological samples were negative.25 »
- Also, why not administer antibiotics for a 7-day period as suggested by ACOGs practice bulletin and extensively studied as beneficial
- According to Guidelines for clinical practice from the French College of Gynaecologists and Obstetricians (CNGOF): “Antibiotic prophylaxis should be prescribed for a period of 7days (Grade C). Nonetheless, as bacterial resistance develops after long treatments, stopping this antibiotic prophylaxis early appears acceptable, even though it has not been assessed in this situation of an initial vaginal sample that turned out to be negative (professional consensus).” (Schmitz et al., 2019 -Eur J Obstet Gynecol Reprod Biol)
- Why were cephalosporins chosen and not erythromycin (or other macrolides) plus GBS prophylaxis, if indicated, with penicillin?
- According to Guidelines for clinical practice from the French College of Gynaecologists and Obstetricians (CNGOF): “Theoretical arguments indicate that amoxicillin (parenteral or oral) or third-generation cephalosporins (parenteral) can each be used alone, but they have not been evaluated for this indication (professional consensus). In older studies of uncertain external validity, given the evolution in bacterial ecology, erythromycin, both with and without amoxicillin (parenteral or oral), showed neonatal benefits (LE1). These substances can therefore be used (professional consensus).” (Schmitz et al., 2019 -Eur J Obstet Gynecol Reprod Biol)
- Please explain your routine clinical practice on the endpoint of conservative management - how, when and who decides on induction
- Our routine clinical practice consisted in expectant management until 37+0W and labour induction was proposed after this gestational age. Earlier induction (or cesarean section) was decided by the obstetrician on call in case of suspected intrauterine infection, bleeding, or non reassuring fetal status.
- Bleeding and preeclampsia, which occur after PPROM, are not baseline characteristics, rather pregnancy outcomes.
- We agree with your comment, we choose to present these pregnancy outcomes in Table 2 « Characteristics of pregnancy and intercurrent events. » to describe them globally.
- « Bleeding » is also presented in Table 4 during the expectant period.
- The length and type of antibiotic treatment, which a proven factor for prolonged latency and may not be similar between tour groups should be presented
- These data were presented in table 4 as “Antibiotic therapy after VS+ or UC+, n (%) ». It corresponded to patients with prolonged antibiotherapy after H48 due to positive bacteriological samples. These data were reported separately during the first and the second week of latency The type of antibiotic tretment was decided in accordance with antibiogram.
- Please define “need for tocolysis” as this in not routinely indicated, and may be a relative contraindication” for PPROM. Who gets tocolysis and for how long, and how does it affect home care
- At admission, tocolysis was mainly initiated before maternal transfer to our institution at physician discretion. This data was reported in table 3. This tocolysis was usually interrupted during the first day after admission or at the latest after a maximal duration of 48 hours. According to our national guidelines « In the absence of any demonstrated neonatal benefits, there is not sufficient evidence to recommend (or to recommend against) initial tocolysis for preterm PROM (Grade C). If tocolysis is prescribed, it should not continue longer than 48 h (Grade C).”(Schmitz et al., 2019 -Eur J Obstet Gynecol Reprod Biol)
- During the expectant period, tocolysis could be re-initiated in case of painful contractions at physician discretion and after ruling out intrauterine infection or non reassuring fetal status. Data were reported in table 4. For outpatients, the “need for tocolysis” was an indication for re-admission.
Submission Date
19 November 2020
Date of this review
12 Dec 2020 07:47:00
